# Potential Effects of Long-Term Exposure to Air Pollution on Dementia: A Longitudinal Analysis in American Indians Aged 55 Years and Older

**DOI:** 10.3390/ijerph21020128

**Published:** 2024-01-24

**Authors:** Yachen Zhu, Yuxi Shi, Scott M. Bartell, Maria M. Corrada, Spero M. Manson, Joan O’Connell, Luohua Jiang

**Affiliations:** 1Program in Public Health, University of California, Irvine, CA 92697, USA; 2Department of Epidemiology and Biostatistics, University of California, Irvine, CA 92697, USAmcorrada@uci.edu (M.M.C.); 3Department of Environmental and Occupational Health, University of California, Irvine, CA 92697, USA; 4Department of Neurology, School of Medicine, University of California, Irvine, CA 92697, USA; 5Centers for American Indian and Alaska Native Health, Colorado School of Public Health, University of Colorado Anschutz Medical Campus, Aurora, CO 80045, USA; spero.manson@cuanschutz.edu (S.M.M.); joan.oconnell@cuanschutz.edu (J.O.)

**Keywords:** PM_2.5_, O_3_, NO_2_, Alzheimer disease and related dementias (ADRD), native Americans

## Abstract

(1) Background: American Indians are disproportionately affected by air pollution, an important risk factor for dementia. However, few studies have investigated the effects of air pollution on the risk of dementia among American Indians. (2) Methods: This retrospective cohort study included a total of 26,871 American Indians who were 55+ years old in 2007, with an average follow-up of 3.67 years. County-level average air pollution data were downloaded from land-use regression models. All-cause dementia was identified using ICD-9 diagnostic codes from the Indian Health Service’s (IHS) National Data Warehouse and related administrative databases. Cox models were employed to examine the association of air pollution with dementia incidence, adjusting for co-exposures and potential confounders. (3) Results: The average PM_2.5_ levels in the IHS counties were lower than those in all US counties, while the mean O_3_ levels in the IHS counties were higher than the US counties. Multivariable Cox regressions revealed a positive association between dementia and county-level O_3_ with a hazard ratio of 1.24 (95% CI: 1.02–1.50) per 1 ppb standardized O_3_. PM_2.5_ and NO_2_ were not associated with dementia risk after adjusting for all covariates. (4) Conclusions: O_3_ is associated with a higher risk of dementia among American Indians.

## 1. Introduction

Air pollution, an important source of premature mortality and morbidity, is unevenly distributed across the United States (US) [1]. In the US, people of color experience greater exposure to air pollution than non-Hispanic Whites [2]. This disparity was found among people at all income levels across states, urban centers, and rural areas [2]. American Indians who live on tribal lands face disproportionate health impacts from air pollution, such as ground-level ozone (O_3_). Ground-level ozone is not emitted directly into the air but is created from chemical reactions between volatile organic compounds (VOCs) and oxides of nitrogen (NOx). They react with sunlight to form O_3_ pollution that is harmful to human health. Cars, power plants, refineries, chemical plants, and other sources, including oil and gas industries that are prevalent on tribal lands, emit such pollutants [3,4]. Other air pollutants detrimental to human health include PM_2.5_ (particles with a diameter of 2.5 μm or less) and nitrogen dioxide (NO_2_), which are more commonly found in large metropolitan areas. Long-term/chronic exposure to PM_2.5_, O_3_, NO_x_, or nitrogen dioxide (NO_2_) can trigger local inflammation and oxidative stress in the brain [5,6], which may contribute to neurodegeneration processes and lead to dementia [7,8].

Epidemiological studies have investigated the association between air pollution and dementia in different populations. However, the results are inconsistent, possibly due to variations in study design, population, air pollutants, source of exposure, exposure level, length of follow-up period, outcome assessment, and/or uncontrolled confounding factors. Although most previous studies found positive associations between incident dementia, dementia hospitalization, or cognitive decline and exposure to PM_2.5_ [9,10,11,12,13,14,15,16], NO_x_/NO_2_ [10,11,14,17,18], or O_3_ [12,19,20,21], some found no or negative associations between such outcomes and PM_2.5_ [20,22], NO_x_/NO_2_ [18], or O_3_ [10,23]. Oudin et al. (2018) [24] found that the association differed according to the source of PM_2.5_. For example, PM_2.5_ from traffic exhaust was associated with higher risk of incident dementia, while PM_2.5_ from residential wood burning was not. In contrast, Tonne et al. (2014) [15] reported that PM_2.5_ from traffic was not associated with cognitive change. Moreover, some studies reported different results for different air pollutants [10,12,14,19,20,23]. For instance, Cerza et al. (2019) [19] found positive associations between O_3_ and dementia hospitalization, yet a negative association between NO_2_ and dementia hospitalization in Rome. Carey et al.’s London-based study (2018) [10] described positive associations of PM_2.5_ and NO_2_ with dementia, yet noted a negative association between O_3_ and dementia. In addition to these studies, Wang et al. (2022) [25] recently documented reduced dementia risk associated with air quality improvement regarding PM_2.5_ and NO_2_ among older women living in the US.

Although numerous studies have examined the association between air pollution and dementia, none have specifically investigated this problem among American Indians. American Indian communities have lower access to and use of health services than non-Hispanic Whites [26] and are disproportionately affected by a higher burden of various chronic diseases [27,28]. To address this gap, we linked individual-level data on dementia diagnosis and related health conditions extracted from the Indian Health Service (IHS) National Data Warehouse with county-level publicly available air pollution and geographic information to investigate the effects of air pollution on dementia risk in American Indians.

## 2. Materials and Methods

### 2.1. Indian Health Service Improving Health Care Delivery Data Project

The IHS funds healthcare services for approximately 2.6 million American Indians and Alaska Natives (AI/ANs) across 574 federally recognized Tribes in 37 states, accounting for about a third of the AI/AN population in the US [29,30]. These services are provided by hospitals, clinics, and health programs operated by the IHS (I), Tribal organizations (T), and urban Indian health programs (U). Known collectively as I/T/Us, most of these programs offer health and health education services that focus on primary and preventive care. They directly provide services when possible; however, when unavailable, AI/ANs may be referred to other providers for specialty care through the Purchased/Referred Care (PRC) program.

We employed the IHS Improving Health Care Delivery Data Project (IHS Data Project) data infrastructure that includes health status and service use information for over 640,000 AI/ANs during fiscal years (FY) 2007–2013, representing about 30% of AI/ANs who use IHS services [31]. This data infrastructure is a synthesis of existing electronic health record data from multiple IHS platforms. It includes data for a purposeful sample of AI/ANs who lived in 15 IHS Service Units (project sites), which are IHS geographic classifications, located throughout the nation. One project site is in the East; four in the Northern Plains; two in the Southern Plains; five in the Southwest; two in the Pacific Coast region; and one in Alaska. These 15 sites had limited urban Indian clinic use during the study period; for that reason, we refer to them as IHS and Tribal (I/T, instead of I/T/U) providers in this study. Data for the current study include registration, demographic, health coverage, and I/T service use data drawn from the IHS National Data Warehouse (NDW) and non-I/T-provided but I/T-paid service use data from the PRC program. More detailed information about this data infrastructure is reported elsewhere [31].

### 2.2. Study Population

The IHS Data Project population is comparable to the national IHS service population in terms of age and gender. For the current study, we restricted our study population to those who were 55 years or older in FY2007 (older adults), used IHS/tribal services at least once each year between FY2007–2009 (defined as the baseline period of this study), registered at same project site during baseline, and, to exclude prevalent cases, were dementia-free at baseline. In other words, we only included those who were regular users of the IHS/tribal services every year and registered at the same site during the baseline period. Meanwhile, we excluded those who had a dementia diagnosis at baseline. Furthermore, we only included those individuals who live in the contiguous US, for whom air pollution data are readily available from the Center for Air, Climate, and Energy Solutions (CACES) [32]. We excluded those from the Alaska site because the CACES group did not include Alaska in their air pollution modeling. We also excluded the counties with ≤5 older adults. Our final analytical data included a total of 26,871 older American Indian adults from 64 counties in the contiguous US.

### 2.3. Measures

#### 2.3.1. Air Pollution Data

We obtained yearly county-level air pollution data in contiguous US counties for three common air pollutants (PM_2.5_, O_3_, and NO_2_) from land-use regression models developed by the CACES [32]. The models were built based on publicly available air pollution concentration measurements from US Environmental Protection Agency (EPA) regulatory monitors, information on land use (e.g., locations of major/minor roads, elevation, and urban/rural area information), and satellite-derived estimates of air pollution to predict concentrations at locations without measurements [32]. While NO_2_ and O_3_ are measured every hour, PM is collected on a daily basis. Model estimates are annual average values for PM_2.5_ and NO_2_ from all sources, such as emissions from the combustion of gasoline, oil, diesel fuel, and wood. For O_3_, the model estimates are the annual warm-season average (during May through September) of the daily maximum 8 h moving average ground-level O_3_. Ground-level O_3_ is more easily formed during the warm season [33], and this metric has long been used in environmental epidemiology to study the long-term health effects of ground-level O_3_ [14,34]. The CACES group provided air pollution prediction at multiple geographic units including county, census tract, and census block group. In this study, we linked the IHS users’ individual-level health data to county-level air pollution data based on their county of residence in FY2007. We averaged the estimated air pollution levels obtained from CACES over the 5-year period before baseline (i.e., from 2003 to 2007), a commonly used length of time [11,14], to represent long-term exposure to air pollution.

#### 2.3.2. Outcome

After excluding those with prevalent dementia at baseline (FY2007–2009), adults were identified as having incident all-cause dementia if they had at least one qualifying ICD-9 diagnostic code in their NDW or PRC inpatient and outpatient service utilization records during the follow-up period (FY2010–2013). The qualifying ICD-9 codes included those for Alzheimer’s disease as well as vascular, Lewy body, frontotemporal, alcohol-induced, and other types of dementia that were used in a recent Medicare study (Appendix A) [35]. The average length of follow-up in this cohort is 3.67 years.

#### 2.3.3. Covariates

Potential confounding variables that might influence the association between dementia and air pollution include individual-level (a) demographics (i.e., age and gender); (b) service region and health coverage in addition to access to IHS/Tribal services (i.e., Medicaid, Medicare, and private insurance); (c) relevant comorbid conditions including depression, diabetes, hypertension, and cardiovascular disease (CVD) at baseline [36,37]; and (d) county-level socioeconomic status (SES). Comorbid chronic conditions were identified using ICD-9 codes reported in the inpatient and outpatient service utilization records in FY2007–2009, supplemented by medication data and some laboratory values. Using a validated algorithm, employed in other national studies and based on diagnostic codes, medication codes, and blood sugar levels [38], we identified diabetes patients among the I/T users. The Sightlines^TM^ DxCG Risk Solutions software (version 4.0.1) groups ICD-9 codes into Diagnostic Cost Groups, which are employed by the federal government and private insurers to identify chronic conditions [39]. We used this software to identify adults with one or more types of comorbidities: CVD, hypertension, and depression. The county-level SES measures were derived from 2010–2014 American Community Survey data from the U.S. Census Bureau [40] for all races in a county. The educational attainment measure was the percentage of adults aged ≥ 25 years who did not complete high school within a county, and the income measure was the percentage of households with incomes below the federal poverty level.

### 2.4. Statistical Analyses

We used Cox proportional hazard regression models to examine the association of air pollution with dementia incidence. Because age is considered the most important risk factor for the development of dementia, we used age as the time scale and accounted for left truncation with truncation time specified as age in FY2007, which allows for non-parametric specification of the age effect and automatically adjusts for the confounding effect of age in the elderly population [41,42]. Based on the previous literature and availability of data in the current study, the potential confounding variables we added to the regression models included (a) gender (Model 1); (b) Model 1 plus project sites (Model 2); (c) Model 2 plus baseline comorbidities, namely depression, diabetes, hypertension, and CVD (Model 3); (d) Model 3 plus health care coverage (Model 4); (e) Model 4 plus county-level SES (Model 5); and (f) Model 5 plus two other air pollutants (Model 6).

We adjusted for the effects of potential confounders in the Cox regressions and computed robust standard errors [43] to account for the fact that between-county variations may be larger than within-county variations in air pollution levels and sociodemographic characteristics. We standardized air pollution levels (i.e., subtract by the mean and then divide by the standard deviation) before running regression models to make the effect estimates comparable across different pollutants. We checked the proportional hazard (PH) assumption by evaluating the independence between Schoenfeld residuals and time [44] and did not find strong evidence for violation of the PH assumption. To ensure study participants were exposed to air pollutants before developing dementia, we conducted a sensitivity analysis using the 5-year average air pollution in 2000–2004 to increase the lag between air pollution exposure and disease outcome. All statistical analyses were performed on R 3.6.2. Data cleaning and management were performed using the tidyverse packages, and survival analyses were performed using the survival package in R.

## 3. Results

Our study included 26,871 American Indians who were 55 years or older in FY2007, registered at the same project site and used IHS services at least once each year between FY2007–2009, and were dementia-free at baseline. Descriptive statistics of individual-level characteristics are shown in Table 1. Among all individuals included in this study, about 54% were in the 55–64 age group, and 59% were female. Almost half of the individuals with dementia were 75 years or older, while only 13% of those without dementia were 75+ years old. Those with dementia had higher percentages of all comorbidities investigated, including depression, diabetes, hypertension, and CVD. More individuals with dementia were enrolled in Medicaid than those without dementia (26.01% vs. 15.27%), while fewer individuals with dementia had private insurance than those without dementia (20.28% vs. 27.15%).

Figure 1 shows the yearly average air pollution levels in all 3109 counties in the contiguous US and in the 64 IHS counties where our study participants resided. The average PM_2.5_ levels in the 64 IHS counties were consistently lower than those in all 3109 counties in the contiguous US during the period from 1999 to 2013, while the NO_2_ levels in the 64 counties were like those of the US. Mean O_3_ levels in the 64 counties were higher than those of all the US counties in most years.

In Table 2, we illustrate adjusted associations between dementia and air pollution from a series of Cox regression models. In the models that adjusted for gender, project sites, comorbidities, health coverage, and county-level SES (Model 5), higher NO_2_ levels were associated with lower risk of dementia (HR = 0.91 and 95% CI: 0.83–0.99), while higher O_3_ levels were associated with higher risk of dementia (HR = 1.28 and 95% CI: 1.11–1.48). There was no association of dementia with PM_2.5_ (HR = 0.85 and 95% CI: 0.66–1.10). However, after adding co-exposure to other air pollutants to the regression model (Model 6), dementia and NO_2_ were not associated with each other (HR = 0.92 and 95% CI: 0.76–1.10), while O_3_ was still strongly associated with a higher risk of dementia (HR = 1.24, 95% CI: 1.02–1.50 per 1 standardize deviation of O_3_). Associations of dementia with PM_2.5_ were not significant in Model 6.

Table 2 also reveals that the relationships between dementia and air pollution differ by gender. While neither PM_2.5_ nor NO_2_ levels were associated with the risk of dementia in either female or male American Indians, as reflected by their wide confidence intervals, higher O_3_ levels were associated with higher risk of dementia among females (HR = 1.39 and 95% CI: 1.04–1.85). However, O_3_ was not associated with dementia risk among American Indian men (HR = 0.98 and 95% CI: 0.64–1.51).

## 4. Discussion

In this study, we investigated the longitudinal associations between three common air pollutants (PM_2.5_, O_3_, and NO_2_) and all-cause dementia in a large American Indian population aged 55 years and older who resided in the contiguous US. Overall, we found that a higher level of O_3_ was associated with higher risk of incident dementia. We also found negative associations between PM_2.5_ and incident dementia in the unadjusted model, but the associations became positive after adjusting for covariates and other air pollutants. No association between NO_2_ and incident dementia was found in any of the regression models except Model 5. Furthermore, we found that the association between O_3_ and dementia risk was substantially higher among female American Indians than that among males. Meanwhile, neither PM_2.5_ nor NO_2_ levels were associated with dementia incidence among either female or male American Indians in gender-stratified analyses.

Our results are consistent with the findings from Cleary et al. (2018) [20], who, in a geographically heterogeneous and broadly distributed sample from the National Alzheimer’s Coordinating Center, observed an increased rate of cognitive decline associated with O_3_, yet not with PM_2.5_. Similarly, Chen and Schwartz (2009) [45] reported consistent associations between a higher level of O_3_ and reduced cognitive performance in US adults. In addition, Cerza et al. (2019) [19] also found a positive association between O_3_ and dementia hospitalization, no association between NO_x_ and dementia hospitalization, but mixed findings between PM_2.5_ and different dementia subtypes (i.e., a positive association with vascular dementia yet a negative association with Alzheimer’s disease).

The effects of air pollution on dementia have been previously explored in numerous studies in the US [13,14,25,46] and other countries around the world [10,11,12,18,19,21,23]. However, none of them investigated this problem among American Indians, some of whom live on or near tribal lands with high ground-level ozone levels [3,4]. To our knowledge, this is the first longitudinal study to investigate the long-term effects of air pollution on dementia incidence among American Indians. Although multiple epidemiological studies have found associations between PM_2.5_ and elevated dementia risk [10,11,12,13,14,23,46], the setting of our study is very different from past work, which were mostly conducted in urban settings. IHS Data Project sites are primarily located in rural areas where the PM_2.5_ levels are much lower than the overall US averages during 1999–2013 (Figure 1). Furthermore, some of the previous studies did not adjust for exposure to other air pollutants, such as O_3_. Although the three air pollutants are not highly correlated with each other, with Pearson’s correlations of 0.14 or less (Appendix B), the adjustment for other air pollutants in the regression models substantially changed some of the parameter estimates, especially for PM_2.5_, suggesting the other air pollutants could be important confounders for the association of each pollutant with dementia.

With respect to gender-specific analyses, in previous studies [47,48,49] that investigated gender differences in the association between air pollution and dementia in high-income countries, most of them found that the associations were stronger among females. Consistent with those studies, we also found strong association between O_3_ and dementia risk among American Indian women, but not in American Indian men. Based on previous studies of AI/ANs, it is not surprising to observe gender differences in the effects of air pollution on dementia risk in this population. For example, it has been reported that AI/AN males were more likely to die before age 65 than AI/AN females, and they were more likely to die due to heart disease and diabetes [50,51]. Thus, early mortality risk might be a competing risk of dementia for American Indian men. In the past, we also found diabetes was cross-sectionally associated with dementia among AI/AN women but not men [37]. Diabetes is another well-established dementia risk factor but was found to be uncorrelated with dementia among AI/AN men, which could also be partially explained by the potential competing risk associated with early mortality in this subgroup of the AI/AN population. Furthermore, substantial biological or physiological distinctions between men and women could also partially explain the gender differences in the association of air pollution and dementia risk observed in the current study.

The strengths of this study include a large, well-characterized cohort of American Indian older patients from geographically diverse sites across the US and the availability of multiple linked data sources with information on diagnosed comorbid conditions, health service utilization, and county-level socioeconomic status, which yield a wide range of potential confounders. Many of the comorbid conditions, such as CVD and diabetes, have been identified as important modifiable risk factors for dementia and thus could be potential confounders for the associations between air pollution and dementia. Some previous studies also suggest that CVD and diabetes could mediate the association between air pollution and dementia, suggesting their inclusion as model covariates might lead to over-adjustment. However, adding these chronic conditions to the models did not materially change the estimates for the parameters of interest (Table 2), indicating the potential robustness of our study findings with respect to chronic conditions. Another strength of this study is the longitudinal study design with long-term air pollution measures available at least 3 years before incident dementia cases, which provides stronger evidence for a causal link than in cross-sectional studies.

This study has several limitations. First, our air pollution exposure was only assessed at the county level because the county code is the smallest standard geographic unit available in the IHS Data Project. County-level exposure estimates may not be the most accurate proxy for long-term environmental exposure for each participant [52]. The exposure aggregation at the county level can lead to effect estimates biased toward the null with less precision (i.e., larger standard errors for effect estimates) compared to fully individual-level studies. This could partially explain the lack of associations between dementia risk and PM_2.5_ and NO_2_ found in the current study. However, it may be helpful to inform environmental policies at the county-level. Moreover, studies that assessed air pollution exposure based solely on individual-level residential address may not be very accurate either by ignoring exposure at the workplace, during regular commutes, or during other social/physical activities [53,54]. Using more detailed residential and work addresses for the IHS users and evaluating the air pollution exposure based on this individual-level information would be a valuable future extension to the current study.

Second, we were only able to adjust for a few major risk factors for dementia at the individual-level including age, gender, and baseline comorbid conditions. Other important potential confounders or effect modifiers are not available, such as apolipoprotein (APOE) genotype and level of physical activity [13,55]. However, Wang et al. (2022) [25] reported that the associations between air pollution and dementia did not substantially differ by APOE genotype among US elderly women. Similarly, previous literature suggested that air pollution may prevent people from engaging in physical activity in highly polluted environments [56]. Therefore, instead of being a potential confounder, physical activity may lie on the causal pathway between air pollution and dementia and, thus, should not be adjusted for in the regression model as a covariate when air pollution is the main exposure. Future studies could investigate the joint effect of air pollution and physical inactivity on dementia or the extent to which physical inactivity explains the association between air pollution and dementia using causal mediation analysis in this population.

Third, identifying dementia patients via clinical diagnostic codes likely underestimates the prevalence and incidence of dementia in the population under consideration. Using 3 years of baseline may not be long enough to capture all the baseline prevalent dementia patients, which means some of the prevalent dementia patients may have been misclassified as incident cases and might lead to biased estimates of the association between air pollution and dementia risk. Meanwhile, an average of 3.67 years of follow-up is also relatively short and we are limited in statistical power because relatively few incident cases of dementia are expected during the few follow-up years given dementia is a chronic disease that usually takes many years to develop. Additionally, the providers at different locations and counties likely use somewhat different criteria when making dementia diagnoses. It is unclear if the potential geographic heterogeneity in the practice of dementia diagnosis could lead to biases in our estimates for the parameters of interests and warrants further study on this topic.

Last, as the study only included those who were regular users of the IHS services every year and registered at the same site during the baseline period, the analytical sample could be biased toward those with more chronic conditions who needed to visit their IHS/tribal providers regularly. As shown in Appendix E, when comparing those included in our study sample (*n* = 26,871) to those who were excluded due to lack of regular use of the IHS/tribal services at the same location but otherwise eligible for this study (*n* = 25,815), we found that the individuals included in our study sample had a similar mean age as those who were excluded, but the included sample had a higher percentage of females (59.5% vs. 54.7%), a higher percentage of Medicare enrollment (56.5% vs. 46.8%), and a higher proportion of private health insurance coverage (33.6% vs. 25.3%). Meanwhile, the included sample had a lower proportion of individuals enrolled in Medicaid than the excluded sample (15.9% vs. 16.8%). The potential selection bias in our study sample again calls for a future study that includes individual addresses for each IHS user to further evaluate the associations between air pollution exposures and dementia risk among American Indians.

## 5. Conclusions

In summary, our study suggests that exposure to O_3_ is associated with higher risk of dementia in American Indians, a population that has long suffered from air pollution and various kinds of chronic diseases. Our findings indicate future efforts to reduce exposure to air pollution may help lower dementia risk in American Indian communities.

## Figures and Tables

**Figure 1 ijerph-21-00128-f001:**
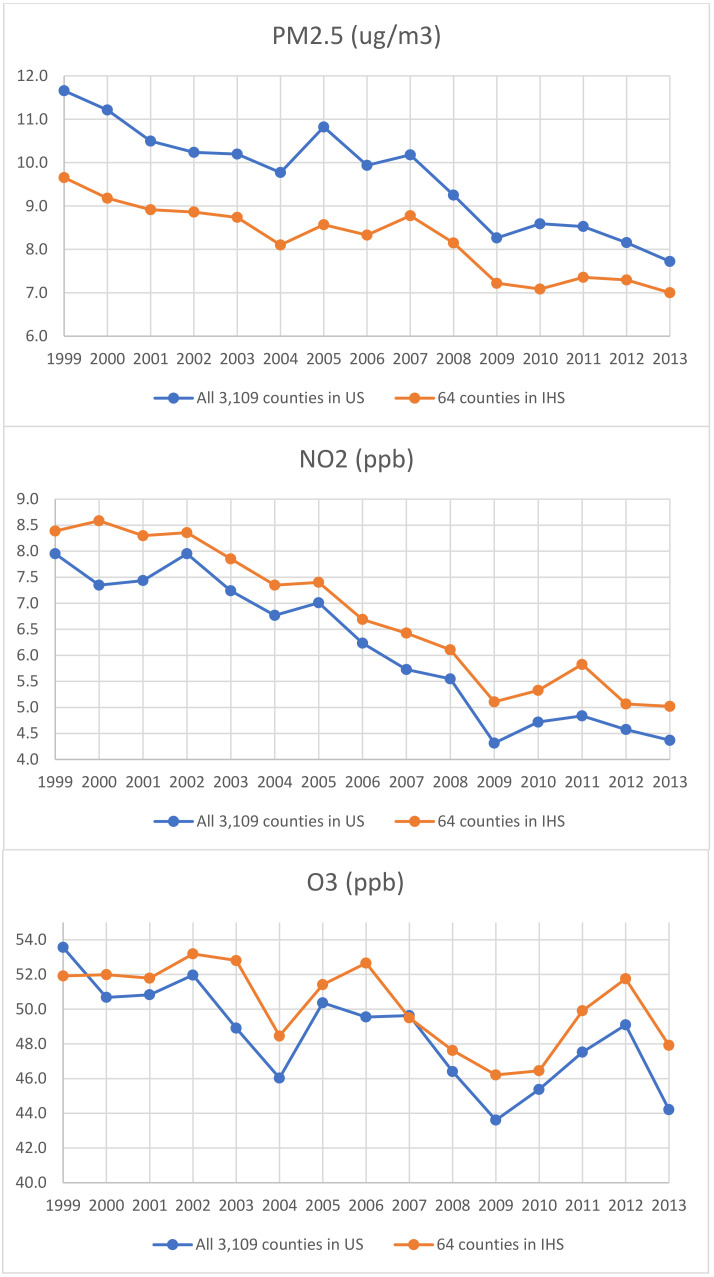
Yearly average air pollution levels of PM_2.5_, NO_2_, and O_3_ in all 3109 US counties versus the 64 counties where the Indian Health Service (IHS) users resided in this study.

**Table 1 ijerph-21-00128-t001:** Descriptive statistics of individual-level characteristics of the study participants.

	All		with Dementia		without Dementia	
	*n* = 26,871		*n* = 1011		*n* = 25,860	
	N	%	N	%	N	%
**Age (FY2007)**						
55–64	14,513	54.01	147	14.54	14,366	55.55
65–74	8561	31.86	374	36.99	8187	31.66
≥75	3797	14.13	490	48.47	3307	12.79
**Gender**						
Female	15,974	59.45	639	63.20	15,335	59.30
Male	10,897	40.55	372	36.80	10,525	40.70
**Region**						
East	1176	4.38	73	7.22	1103	4.27
Northern Plains	2767	10.30	119	11.77	2648	10.24
Pacific Coast	1282	4.77	32	3.17	1250	4.83
Southern Plains	12,066	44.90	308	30.46	11,758	45.47
Southwest	9580	35.65	479	47.38	9101	35.19
**Baseline depression**	5418	20.16	261	25.82	5157	19.94
**Baseline diabetes**	12,690	47.23	501	49.55	12,189	47.13
**Baseline hypertension**	21,124	78.61	853	84.37	20,271	78.39
**Baseline CVD**	10,868	40.45	566	55.98	10,302	39.84
**Medicaid (FY2009)**	4212	15.67	263	26.01	3949	15.27
**Private health coverage (FY2009)**	7227	26.90	205	20.28	7022	27.15

**Table 2 ijerph-21-00128-t002:** Adjusted hazard ratios (HRs) of air pollution levels for dementia risk (HR per 1-unit increase in standardized air pollution levels, 95% CI).

	PM_2.5_	NO_2_	O_3_
	HR (95% CI)	HR (95% CI)	HR (95% CI)
Overall Sample (*n* = 26,871)			
Model 1	0.80 (0.73–0.88) ***	0.98 (0.87–1.11)	1.08 (0.96–1.22)
Model 2	0.80 (0.60–1.08)	0.91 (0.80–1.03)	1.18 (0.90–1.55)
Model 3	0.83 (0.63–1.10)	0.92 (0.82–1.03)	1.18 (0.93–1.50)
Model 4	0.85 (0.65–1.12)	0.93 (0.83–1.04)	1.19 (0.95–1.50)
Model 5	0.85 (0.66–1.10)	0.91 (0.83–0.99) *	1.28 (1.11–1.48) ***
Model 6	1.12 (0.69–1.83)	0.92 (0.76–1.10)	1.24 (1.02–1.50) *
Females (*n* = 15,974)			
Model 1	0.81 (0.73–0.91) ***	1.00 (0.89–1.13)	1.15 (1.02–1.29) *
Model 2	0.93 (0.55–1.59)	0.91 (0.75–1.09)	1.25 (0.79–1.97)
Model 3	0.97 (0.58–1.61)	0.92 (0.77–1.10)	1.25 (0.82–1.91)
Model 4	0.97 (0.58–1.62)	0.92 (0.77–1.11)	1.27 (0.83–1.94)
Model 5	1.00 (0.64–1.55)	0.90 (0.79–1.03)	1.39 (1.07–1.79) *
Model 6	1.76 (0.94–3.31)	0.81 (0.63–1.05)	1.39 (1.04–1.85) *
Males (*n* = 10,897)			
Model 1	0.78 (0.69–0.88) ***	0.95 (0.83–1.10)	0.99 (0.86–1.14)
Model 2	0.59 (0.30–1.15)	0.90 (0.72–1.12)	1.08 (0.76–1.54)
Model 3	0.61 (0.31–1.24)	0.91 (0.73–1.13)	1.10 (0.76–1.57)
Model 4	0.64 (0.32–1.29)	0.92 (0.74–1.14)	1.10 (0.76–1.58)
Model 5	0.60 (0.29–1.24)	0.90 (0.73–1.12)	1.12 (0.76–1.65)
Model 6	0.52 (0.18–1.55)	1.07 (0.77–1.50)	0.98 (0.64–1.51)

Model 1 adjusted for gender. Model 2 further adjusted for project sites. Model 3 further adjusted for baseline depression, diabetes, hypertension, and CVD. Model 4 further adjusted for healthcare coverage (Medicaid and private insurance). Model 5 further adjusted for county-level % poverty and % education below high school completion in all races. Model 6 further adjusted for the other two air pollutants. * *p*-value < 0.05, *** *p*-value < 0.001.

## Data Availability

The data from the IHS Data Project used to support the findings of this study have not been made available because of IHS and Tribal regulations regarding data confidentiality and security.

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
