# Peer review of "Potential Effects of Long-Term Exposure to Air Pollution on Dementia: A Longitudinal Analysis in American Indians Aged 55 Years and Older"

_ijerph, 2024, doi:10.3390/ijerph21020128_

Round 1
Reviewer 1 Report
Comments and Suggestions for Authors
This cohort study investigated air pollution (NO2, PM10, PM2.5 and O3) associations at the county-level with incident dementia of any type. Overall, this is an interesting paper in terms of the population it chose to study – American Indians - and also that these communities are mostly rural. The authors appropriately acknowledge some major limitations such as the rather crude spatial resolution of their air pollution estimation (county-level) and the potential to underestimates the prevalence and incidence of dementia due to using clinic ICD codes only and the limited number of covariates available at the individual level. There are, however, additional issues that need to be addressed particularly the use of an all dementia category only (instead of stratifying by dementia type) and a more extensive exploration of the results in sensitivity analyses (such as stratifications by gender and neighborhood SES). Such additional information would help the reader to better understand the results and improve the overall quality of the manuscript. Finally, a better interpretation of the sources for these three air pollutants and their general behavior (spatial distribution) would also help the reader to better understand the results especially the ones that otherwise might seem counterintuitive (such as the negative associations estimated for NO2 and PM2.5).
Please, answer to the following comments:
· Abstract: please add how the outcome (dementia) was assessed/defined and name the study design clearly (record-based cohort study?) including the number of years of follow-up etc.
· Please state clearly the average length of follow-up in this cohort and discuss the limitations of the relatively short FU in the discussion.The authors mentioned that they used the clinical records during 2010-2013 to identify the outcome dementia,, but it is not entirely clear whether or how they employed the records for 2007-2009. The authors assumed that the cases they identified in 2010-13 were incident cases based on the fact that they excluded any cases of dementia at baseline 2007-2009? Is it correct that someone who is not recorded with dementia in year 2007 and 2008 but in the year 2009 would thus be excluded, as the requirement is to have 3 consecutive years without a dementia diagnosis? Please clarify this. Also, please clarify, when does the follow-up time begin to count for this cohort, in the year 2007 or the year 2010? It seems that 2007-09 is possibly an immortal time period during which no subject was allowed to develop dementia or die or move out of the county or be not seen in the IHS system (as the authors state:” we restricted our study population to those who …, used IHS services at least once each year between FY2007-2009 (defined as baseline of this study), registered at same project site during baseline, and, to exclude prevalent cases, were dementia-free at baseline. “ ?
· The baseline selection of individuals who had at least one visit in each of 3 years with the IHS clearly favored individuals with chronic diseases such as diabetes, CVD, and hypertension as can be seen in table 1 – thus this is a cohort of individuals highly vulnerable to dementia. This should be clearly discussed in terms of its implications. It might also help to show how those individuals who only saw the IHS once were similar or different from those eligible for this study or all individuals who may or may not have used the IHS in these years but were eligible to do so?
· For the ICD code used for the outcome diagnosis, please specify whether the authors only considered the primary diagnosis from the records, or including the 2nd, 3rd diagnosis? How was dementia clinically diagnosed? Is anything known about what types of examination or criteria the clinics used to diagnose dementia?
· The authors mentioned in line 82-84 “I/T/Us directly provide services when possible; however, when unavailable, AI/ANs may be referred to other providers for specialty care through the Purchased/Referred Care (PRC) program.” What difference does this make for the dementia diagnoses? Would these providers use the same criteria? Or is one or another provider less likely to diagnose dementia, and would this be different by county? Please clarify.
· For the land-use regression models developed by the CACES, it would help if the authors provided a little more details and not just cited a previous paper, especially to understand better what the sources of PM2.5 were that they modeled?
· Please present and discuss correlations between the three air pollutants.
· The authors used mean standardization of pollutants in the analyses to make them comparable. However, in air pollution epidemiology instead using an per IQR increase is more common, easier to interpret, and it will allow the reader to compare effect sizes across studies more easily (after converting the scale to the actual ppb etc). I suggest to provide the IQRs and also consider showing in a supplement the HRs per original unit increase.
· Please provide a more thorough discussion and interpretation of the multi-pollutant models. When you adjusted the PM2.5 model for NO2 and O3, the direction of the association changed (Table 2), please explain this. How would the authors interpret for this change, for example, would NO2 capture the variability in the outcome due to traffic emissions and PM2.5 non-traffic? Please clarify.
· Also, as NO2 and ozone are generally mirror images of each other at smaller geospatial scale due to atmospheric chemistry (with NO2 higher close to traffic and ozone found away from traffic), how do the authors interpret to opposite side of the null effect estimates? I recommend examining and reporting the correlation structure of the air pollution measures used here. It is also well-known that ozone and PM2.5 are much more spatially homogeneously distributed than NO2. Did the authors consider this?
· I understand that the study did not include some covariates such as physical activity as the authors mentioned, however, how would physical activity be associated with county level air pollution in order to act as a potential confounder? I.e. would physical activity be expected to be different between more or less rural versus urban counties?
· It is unclear whether the participants were allowed to move during the study period from one tribal area county to another or whether everyone included was required to be a non-mover. How does this distinguish the participants from Native Americans who moved and were not included in the analyses?
· Please conduct and provide information on some sensitivity analyses including analyses stratified by gender, by US region (or at least restricted to the Southwest and Southern Plains regions where almost 80 % of these counties are located), and by age 65+.
· Please provide also some sensitivity analyses by specific type of dementia (granted that most dementia cases might be of a mixed type – at least at advanced ages), at least for the most common ones such as vascular and AD. If this were possible, the paper would be much more informative overall.
· Please add some descriptive information on county-level variables such as % educational attainment, income, and SES
· I highly recommend to NOT use statistical significance to interpret the results, as this is not state-of the science for epidemiologic studies. Please consider 95%CIs and their overlap with each other or the null value as well as changes in effect estimates etc instead as more meaningful parameters to interpret and base conclusions on.
Comments on the Quality of English Language
no concerns
Author Response
Reviewer: 1
This cohort study investigated air pollution (NO2, PM10, PM2.5 and O3) associations at the county-level with incident dementia of any type. Overall, this is an interesting paper in terms of the population it chose to study – American Indians - and also that these communities are mostly rural. The authors appropriately acknowledge some major limitations such as the rather crude spatial resolution of their air pollution estimation (county-level) and the potential to underestimates the prevalence and incidence of dementia due to using clinic ICD codes only and the limited number of covariates available at the individual level. There are, however, additional issues that need to be addressed particularly the use of an all dementia category only (instead of stratifying by dementia type) and a more extensive exploration of the results in sensitivity analyses (such as stratifications by gender and neighborhood SES). Such additional information would help the reader to better understand the results and improve the overall quality of the manuscript. Finally, a better interpretation of the sources for these three air pollutants and their general behavior (spatial distribution) would also help the reader to better understand the results especially the ones that otherwise might seem counterintuitive (such as the negative associations estimated for NO2 and PM2.5).
Please, answer to the following comments:
- Abstract: please add how the outcome (dementia) was assessed/defined and name the study design clearly (record-based cohort study?) including the number of years of follow-up etc.
Response: The abstract has been revised as suggested.
- Please state clearly the average length of follow-up in this cohort and discuss the limitations of the relatively short FU in the discussion. The authors mentioned that they used the clinical records during 2010-2013 to identify the outcome dementia,, but it is not entirely clear whether or how they employed the records for 2007-2009. The authors assumed that the cases they identified in 2010-13 were incident cases based on the fact that they excluded any cases of dementia at baseline 2007-2009? Is it correct that someone who is not recorded with dementia in year 2007 and 2008 but in the year 2009 would thus be excluded, as the requirement is to have 3 consecutive years without a dementia diagnosis? Please clarify this. Also, please clarify, when does the follow-up time begin to count for this cohort, in the year 2007 or the year 2010? It seems that 2007-09 is possibly an immortal time period during which no subject was allowed to develop dementia or die or move out of the county or be not seen in the IHS system (as the authors state:” we restricted our study population to those who …, used IHS services at least once each year between FY2007-2009 (defined as baseline of this study), registered at same project site during baseline, and, to exclude prevalent cases, were dementia-free at baseline. “ ?
Response: The average length of follow-up in this cohort is 3.67 years. The baseline period is defined as FY2007 - FY2009, and the follow-up period is defined as FY2010 - FY2013. We excluded those with a dementia diagnosis during the three years of the baseline period in order to identify incident dementia cases during the follow-up period. And based on this definition, yes, if someone who is not recorded with dementia in FY2007 and FY2008, but with dementia diagnosis in FY2009, this person will be excluded since we require the analytical sample only incudes those who were dementia free during 3-year baseline period. We have clarified these points in the Methods section, Study Population and Outcome subsections. We also added a few sentences to the limitations section acknowledging and discussing the relatively short follow-up period:
“Meanwhile, an average of 3.67 years of follow-up is also relatively short and we are limited in statistical power because relatively few incident cases of dementia are expected during the few follow-up years given dementia is a chronic disease that usually takes many years to develop.”
- The baseline selection of individuals who had at least one visit in each of 3 years with the IHS clearly favored individuals with chronic diseases such as diabetes, CVD, and hypertension as can be seen in table 1 – thus this is a cohort of individuals highly vulnerable to dementia. This should be clearly discussed in terms of its implications. It might also help to show how those individuals who only saw the IHS once were similar or different from those eligible for this study or all individuals who may or may not have used the IHS in these years but were eligible to do so?
Response: We agree that the inclusion requirement of having at least one visit at the same project site in each of the three-year baseline period could induce some selection bias by including a higher percentage of individuals with chronic diseases. However, with this requirement, we were able to include those who were more likely to live in the same location/county over the study period, which is important to examine the effects of air pollution on health outcomes using county level air pollution data. We have added this as one of the limitations to the Discussion section.
In addition, as suggested, we compared those who were included in our study sample (n=26,871) and those who were excluded due to lack of regular use of the IHS/tribal services at the same location but otherwise eligible for this study (n=25,815). We found people included in our study sample had similar mean age as those who were excluded (65.6 vs 65.3). Moreover, compared to the excluded sample, the included sample had higher percentage of females (59.5% vs. 54.7%), higher percentage of Medicare enrollment (56.5% vs. 46.8%), and higher proportion of private health insurance coverage (33.6% vs. 25.3%). Meanwhile, the included sample had lower percentage of individuals enrolled in Medicaid than the excluded sample (15.9% vs. 16.8%). Since many of the excluded individuals did not use the IHS/tribal services much, we cannot provide the chronic diseases information for those people.
- For the ICD code used for the outcome diagnosis, please specify whether the authors only considered the primary diagnosis from the records, or including the 2nd, 3rddiagnosis? How was dementia clinically diagnosed? Is anything known about what types of examination or criteria the clinics used to diagnose dementia?
Response: Individuals were identified as having incident all-cause dementia if they had at least one qualifying ICD-9 diagnostic code in any of their encounters available during the follow-up period (FY2010 – FY2013), regardless of the order of their diagnoses.
Unfortunately, we do not have detailed information regarding the types of examinations or the criteria used by the clinicians for dementia diagnosis.
- The authors mentioned in line 82-84 “I/T/Us directly provide services when possible; however, when unavailable, AI/ANs may be referred to other providers for specialty care through the Purchased/Referred Care (PRC) program.” What difference does this make for the dementia diagnoses? Would these providers use the same criteria? Or is one or another provider less likely to diagnose dementia, and would this be different by county? Please clarify.
Response: The providers at different locations likely use somewhat different criteria when making dementia diagnoses. Unfortunately, we do not have detailed information for the providers participating in the PRC program and thus do not know how different the diagnosis criteria could be across different sites, programs and counties. We have added a few sentences to the limitation section, further acknowledging this as an important limitation of our study.
- For the land-use regression models developed by the CACES, it would help if the authors provided a little more details and not just cited a previous paper, especially to understand better what the sources of PM2.5 were that they modeled?
Response: As described in Air Pollution Data section, “The models were built based on publicly available air pollution concentration measurements from US Environmental Protection Agency (EPA) regulatory monitors, information on land use (e.g., locations of major/minor roads, elevation, urban/rural area information), and satellite-derived estimates of air pollution to predict concentrations at locations without measurements [32].” Thus, the PM2.5 modelled could come from all sources, such as emissions from combustion of gasoline, oil, diesel fuel and wood. We have added a few sentences to that section to describe the land-use regression models in more details.
- Please present and discuss correlations between the three air pollutants.
Response: The correlations among the three air pollutants are presented in Appendix 2 now. We also added a few sentences to the Discussion section discussing the correlations and potential confounding effects among the three air pollutants. We also added a few sentences to the Discussion section discussing the correlations and potential confounding effects among the three air pollutants: “Although the three air pollutants are not highly correlated with each other with Pearson’s correlations of 0.14 or less (Appendix 2), the adjustment of other air pollutants in the regression models changed some of the parameter estimates substantially, especially for PM2.5, suggesting the other air pollutants could be important confounders for the association of each pollutant vs. dementia.”
- The authors used mean standardization of pollutants in the analyses to make them comparable. However, in air pollution epidemiology instead using an per IQR increase is more common, easier to interpret, and it will allow the reader to compare effect sizes across studies more easily (after converting the scale to the actual ppb etc). I suggest to provide the IQRs and also consider showing in a supplement the HRs per original unit increase.
Response: As suggested, the IQRs of the air pollutants are provided in Appendix 3. We also presented the HRs per original unit increase in Appendix 4.
- Please provide a more thorough discussion and interpretation of the multi-pollutant models. When you adjusted the PM2.5 model for NO2 and O3, the direction of the association changed (Table 2), please explain this. How would the authors interpret for this change, for example, would NO2 capture the variability in the outcome due to traffic emissions and PM2.5 non-traffic? Please clarify.
Response: This change could be caused by the confounding effects of O3. In descriptive statistics, we found that O3 levels were higher in IHS counties with more dementia cases, while the level of PM2.5 went the opposite direction, with higher levels of PM2.5 found in counties with fewer dementia cases. Therefore, we would expect to observe a positive association between O3 and dementia risk, yet a negative association between PM2.5 and dementia risk without adjusting for the other air pollutants. If the association between PM2.5 and dementia was confounded by O3, we would expect the association between PM2.5 and dementia to change substantially after adding O3 into the model, and it’s possible that the direction of the association also changes. We added a few sentences to the Discussion section discussing the correlations and potential confounding effects among the three air pollutants: “Although the three air pollutants are not highly correlated with each other with Pearson’s correlations of 0.14 or less (Appendix 2), the adjustment of other air pollutants in the regression models changed some of the parameter estimates substantially, especially for PM2.5, suggesting the other air pollutants could be important confounders for the association of each pollutant vs. dementia.”
- Also, as NO2 and ozone are generally mirror images of each other at smaller geospatial scale due to atmospheric chemistry (with NO2 higher close to traffic and ozone found away from traffic), how do the authors interpret to opposite side of the null effect estimates? I recommend examining and reporting the correlation structure of the air pollution measures used here. It is also well-known that ozone and PM2.5 are much more spatially homogeneously distributed than NO2. Did the authors consider this?
Response: As the reviewer pointed out, at smaller geospatial scale, “NO2 was higher close to traffic while ozone found away from traffic”, which is consistent with our findings. In this study, we observed more dementia cases in IHS counties with higher O3 but lower NO2, suggesting opposite directions. This also aligns with the fact that IHS Data Project sites are primarily located in rural areas where air pollution due to traffic is assumed to be lower. Pls see our response to Comment #7 above, indicating we added a few sentences to the Discussion section discussing the correlations and potential confounding effects among the three air pollutants.
- I understand that the study did not include some covariates such as physical activity as the authors mentioned, however, how would physical activity be associated with county level air pollution in order to act as a potential confounder? I.e. would physical activity be expected to be different between more or less rural versus urban counties?
Response: As we acknowledged as a limitation in the Discussion section, we were not able to adjust for physical activity, which is a modifiable risk factor for dementia (Livingston et al., 2020). However, the exiting evidence for the association between physical activity and county level air pollution is inconsistent. A study by Martin et al. (2005) found higher physical activity levels in urban areas than in rural areas nationally and regionally in the South, which could not be generalized to all regions in the US. A more recent study by Robertson et al. (2018) found no difference in total physical activity between urban and rural areas, however, rural residents may engage in less leisure-time physical activity than their urban counterparts. We have added the following sentences to the limitations section to further discuss this issue:
“Additionally, previous literature suggested that air pollution may prevent people from engaging in physical activity in highly polluted environments (Tainio et al., 2021). Therefore, instead of being a potential confounder, physical activity might lie on the causal pathway between air pollution and dementia, thus should not be adjusted for in the model as a covariate when air pollution is the main exposure. Future studies could investigate the joint effect of air pollution and physical inactivity on dementia or the extent to which physical inactivity explains the association between air pollution and dementia using causal mediation analysis in this population.”
- It is unclear whether the participants were allowed to move during the study period from one tribal area county to another or whether everyone included was required to be a non-mover. How does this distinguish the participants from Native Americans who moved and were not included in the analyses?
Response: As we explained in response to a previous related comment, individuals included in our study sample must be registered at same research site during three-year baseline period (FY2010 – FY2013), but they may move during the four-year follow-up period. However, among the included individuals, only 4.95% of them had moved during the four-year follow-up time. The differences between the included and excluded individuals are presented in Appendix 2 (please see our response to Comment #3 above).
- Please conduct and provide information on some sensitivity analyses including analyses stratified by gender, by US region (or at least restricted to the Southwest and Southern Plains regions where almost 80 % of these counties are located), and by age 65+.
Response: Thank you for this great suggestion. We conducted stratified analysis as suggested and found gender is indeed a significant effect modifier for the association between air pollution and dementia risk. Specifically, we found the association between O3 and dementia risk was much stronger among female American Indian (AI) adults than that in male AI adults. In the final model adjusting for all covariates and the other air pollutants, O3 was significantly associated with a increased risk of dementia (HR = 1.39) among females, while the association between O3 and dementia risk was not statistically significant among males. We have added these important findings to Table 2 and the Results section, as well as relevant discussions of these findings to the Discussion section.
We also conducted sensitivity analysis stratified by US region (Southwest and Southern Plains vs. other regions) and found the associations between air pollution and dementia were generally consistent across the two regional strata.
- Please provide also some sensitivity analyses by specific type of dementia (granted that most dementia cases might be of a mixed type – at least at advanced ages), at least for the most common ones such as vascular and AD. If this were possible, the paper would be much more informative overall.
Response: We agree our paper could be much strengthened if we can conduct sensitivity analyses for each specific subtype of dementia. Unfortunately, most incident dementia cases (>80%) identified in this study were given an ICD-9 code for Not Otherwise Specified dementia, likely due to the lack of specialists in the IHS/Tribal health care system at the time of diagnosis. Thus, it is not very informative to perform this type of sensitivity analyses using our current data.
- Please add some descriptive information on county-level variables such as % educational attainment, income, and SES
Response: As suggested, the descriptive statistics of the county level variables are provided in Appendix 3.
- I highly recommend to NOT use statistical significance to interpret the results, as this is not state-of the science for epidemiologic studies. Please consider 95%CIs and their overlap with each other or the null value as well as changes in effect estimates etc instead as more meaningful parameters to interpret and base conclusions on.
Response: As recommended, we always use 95%CIs and their overlap with the null value as well as changes in effect estimates as the primary way to interpret the results and make conclusions. We have removed the words “statistically significant” from the Results and Discussions sections to avoid potentially misleading interpretations.
Reviewer 2 Report
Comments and Suggestions for Authors
Dear Authors,
Your manuscript: „Potential Effects of Long-term Exposure to Air Pollution on Dementia: A Longitudinal Analysis in American Indians Aged 55 years and Older” addresses an important and underexplored topic—the potential effects of long-term exposure to air pollution on dementia in American Indians aged 55 years and older. The study employs a robust longitudinal analysis, examining county-level air pollution data and dementia incidence among a substantial cohort of American Indians. The research is well-structured and presents compelling findings.
The study addresses a significant research gap by focusing on the impact of air pollution on dementia within the American Indian population. This novel approach contributes to the broader understanding of environmental risk factors for cognitive decline.
The use of Cox proportional hazard models and the inclusion of relevant covariates in the analysis demonstrate methodological rigor. Adjusting for co-exposures and potential confounders enhances the study's internal validity.
The presentation of results is clear, with concise reporting of average air pollution levels in Indian Health Service (IHS) counties and the subsequent analysis of the association between ozone (O3) levels and dementia incidence. The inclusion of hazard ratios adds quantitative precision to the findings.
However, minor aspects warrant attention for further clarification.
Areas for Improvement:
Discussion of Non-significant Associations:
First, the manuscript effectively communicates the significant positive association between dementia and O3 levels. However, the discussion could benefit from a more comprehensive exploration of non-significant associations with PM2.5 and NO2. Providing insights into why these associations were not statistically significant would enhance the interpretation of the results.
Second, the discussion could benefit from broader insights into baseline comorbid conditions and their role in the potential risk of dementia. Although some papers suggest that vascular factors could mediate an association between air pollutants and dementia. In your analysis, CVD risk may play a role.
Overall Recommendation:
The manuscript presents a significant contribution to the literature on air pollution and dementia risk among American Indians. To strengthen the study, I recommend addressing the mentioned points for improvement. With these refinements, the manuscript holds the potential for publication in the International Journal of Environmental Research and Public Health.
Author Response
Reviewer: 2
Areas for Improvement:
Discussion of Non-significant Associations:
- First, the manuscript effectively communicates the significant positive association between dementia and O3 levels. However, the discussion could benefit from a more comprehensive exploration of non-significant associations with PM2.5 and NO2. Providing insights into why these associations were not statistically significant would enhance the interpretation of the results.
Response: Thank you for this suggestion. We have added more discussions regarding the potential reasons for the non-significant associations between dementia risk and PM2.5 and NO2 that were found in our study:
“Although the three air pollutants are not highly correlated with each other with Pearson’s correlations of 0.14 or less (Appendix 2), the adjustment of other air pollutants in the regression models changed some of the parameter estimates substantially, especially for PM2.5, suggesting the other air pollutants could be important confounders for the association of each pollutant vs. dementia.”
“The exposure aggregation at county level can lead to effect estimates biased toward the null with less precision (i.e., larger standard errors for effect estimates) compared to fully individual-level studies. This could partially explain the lack of significant associations of dementia risk with PM2.5, and NO2 found in the current study.”
- Second, the discussion could benefit from broader insights into baseline comorbid conditions and their role in the potential risk of dementia. Although some papers suggest that vascular factors could mediate an association between air pollutants and dementia. In your analysis, CVD risk may play a role.
Response: As suggested, we have added a few sentences to the Discussion section discussing baseline comorbid conditions and their potential role in estimating the parameters of interest:
“Many of the comorbid conditions, such as cardiovascular disease and diabetes, have been identified as important modifiable risk factors for dementia and thus could be potential confounders for the associations between air pollution and dementia. Some previous studies also suggest that CVD and diabetes could mediate the association between air pollution and dementia, suggesting including them as model covariates might lead to over-adjustment. However, adding those chronic conditions to the models did not materially change the estimates for the parameters of interest (Table 2), suggesting the potential robustness of the study findings with respect to chronic conditions.”
Round 2
Reviewer 1 Report
Comments and Suggestions for Authors
the authors have answered to the reviewer's suggestions sufficiently. Here are a few additional edits I recommend. Again, please do not use statement such as "lack of significant associations" on page 10 and more generally "statistical significance of an association" as an argument.
-in the figure IHS is misspelled as HIS
- in the new table 2 the total N of the first part is wrong (Overall Sample (n = 10897)
Comments on the Quality of English Languageplease check
Author Response
the authors have answered to the reviewer's suggestions sufficiently. Here are a few additional edits I recommend.
- Again, please do not use statement such as "lack of significant associations" on page 10 and more generally "statistical significance of an association" as an argument.
Response: Thank you for this suggestion. We have removed “significant” from that sentence and revised a few more sentences to avoid using statistical significance as an argument.
- in the figure IHS is misspelled as HIS
Response: Thank you very much for catching this typo. We have fixed it.
- in the new table 2 the total N of the first part is wrong (Overall Sample (n = 10897)
Response: Thank you very much for catching this typo. We have fixed it.
- Minor editing of English language required.
Response: We have asked the Publications Manager and Media Coordinator (Ms. Sara Mumby) at Centers for American Indian & Alaska Native Health to read and check the English language of this manuscript and accepted all of her suggested edits (tracked).